# Collectively enhanced Ramsey readout by cavity sub- to superradiant transition

Eliot A. Bohr [1,4] ✉, Sofus L. Kristensen[1,4], Christoph Hotter [2], Stefan A. Schäffer [1], Julian Robinson-Tait[1], Jan W. Thomsen[1], Tanya Zelevinsky[3], Helmut Ritsch [2] & Jörg H. Müller[1]

When an inverted ensemble of atoms is tightly packed on the scale of its emission wavelength or when the atoms are collectively strongly coupled to a single cavity mode, their dipoles will align and decay rapidly via a superradiant burst. However, a spread-out dipole phase distribution theory predicts a required minimum threshold of atomic excitation for superradiance to occur. Here we experimentally confirm this predicted threshold for superradiant emission on a narrow optical transition when exciting the atoms transversely and show how to take advantage of the resulting sub- to superradiant transition. A $\pi/2$-pulse places the atoms in a subradiant state, protected from collective cavity decay, which we exploit during the free evolution period in a corresponding Ramsey pulse sequence. The final excited state population is read out via superradiant emission from the inverted atomic ensemble after a second $\pi/2$-pulse, and with minimal heating this allows for multiple Ramsey sequences within one experimental cycle. Our scheme is an innovative approach to atomic state readout characterized by its speed, simplicity, and highly directional emission of signal photons. It demonstrates the potential of sensors using collective effects in cavity-coupled quantum emitters.

The precise measurement and manipulation of atomic states lie at the heart of a broad range of scientific and technological advancements. Quantum sensors such as atomic clocks, with their unprecedented accuracy and stability[1–6], have revolutionized fields ranging from fundamental physics to global positioning systems. The development of novel state detection methods plays a pivotal role in improving the performance and capabilities of these atomic sensors[7–12]. In recent years, quantum technologies have emerged as a promising avenue for enhancing sensitivities in measurement and sensing applications. Among these technologies, ultracold atomic systems, combined with high-finesse optical cavities, have demonstrated remarkable potential due to their ability to access and manipulate collective quantum states[13–20].

Ramsey spectroscopy[21], widely used for precision measurements[22,23] in atomic systems, involves a two-pulse sequence that measures the accumulated phase difference between a laser and a superposition of atomic states. By incorporating an optical cavity into the experimental setup, we harness the collective nature of atoms with randomly distributed atom-cavity couplings to generate a state-selective superradiant decay into a well-defined cavity mode. This preferred directional emission allows for efficient collection of nearly all the signal photons compared to more straightforward atomic state detection in which most of the photons are lost into free space. Furthermore, we do not require any additional lasers as in electron shelving, since our readout relies on detecting photons directly from the clock interrogation. We demonstrate the reusability of our atomic ensemble by turning on cooling light between successive Ramsey sequences and show 100s of repetitions on the same atomic cloud, ultimately limited by background gas collisions of the vacuum chamber. Our experimental system comprises ultracold strontium atoms trapped within a high-finesse optical cavity, in the so-

[1]Niels Bohr Institute, University of Copenhagen, Blegdamsvej 17, Copenhagen DK-2100, Denmark. [2]Institut für Theoretische Physik, Universität Innsbruck, Technikerstr. 21a, Innsbruck A-6020, Austria. [3]Department of Physics, Columbia University, 538 West 120th Street, New York 10027-5255 NY, USA. [4]These authors contributed equally: Eliot A. Bohr, Sofus L. Kristensen. ✉e-mail: eliot.bohr@nbi.ku.dk

called bad-cavity regime. The coherent excitation and de-excitation of the atomic ensemble, induced by carefully timed optical pulses, generate an enhanced superradiant emission into the cavity mode when the inversion exceeds threshold. Below this threshold, the atoms are predicted to exhibit suppressed emission, or subradiance[24–32], with respect to the cavity mode[33]. Emission into free-space remains unchanged, and the atoms are still susceptible to single-atom spontaneous decay into modes outside of the cavity's solid angle. This collective emission serves as a sensitive probe for detecting and characterizing the atomic states, enabling improved state discrimination and measurement precision. In addition, the subradiant behavior with respect to the cavity mode allows for interaction times without collective cavity decay, necessary for resolving and exploiting ultranarrow clock transitions.

The use of ultracold strontium atoms in conjunction with an optical cavity presents several advantages for our proposed state detection method. The long coherence times and precise control achievable in ultracold atomic systems facilitate the generation of long-lived population oscillations. The optical cavity provides enhanced light-matter interaction, amplifying the superradiant emission and enabling efficient state readout.

In this article, we present the experimental realization of a distinctive state detection method[33] that exploits the phenomenon of superradiant (SR) light emission[34,35] after transverse Ramsey interrogation of atoms in an optical cavity. We experimentally demonstrate the excitation threshold for superradiant states and employ the interplay of sub- and superradiance for a fast cavity-assisted readout.

## Results
### Experimental implementation
We cool a cloud of up to $N = 40(4) \times 10^6$ $^{88}$Sr atoms down to $2\,\mu$K using a two-stage magneto-optical trap (MOT). The cloud of atoms is then centered in the fundamental mode of an optical cavity (Fig. 1) with finesse $\mathcal{F} = 1001(5)$ and cavity decay rate $\kappa/2\pi = 780(4)$ kHz. By locking the cavity to a reference laser one free spectral range away, we tune a TEM$_{00}$ resonance of the cavity to the $|e\rangle = {}^3P_1$, $m_j = 0 \leftrightarrow |g\rangle = {}^1S_0$ transition frequency, $\omega_a$. The cavity mode waist radius is $450\,\mu$m, and the atomic cloud has a vertical full height of $100\,\mu$m and horizontal full width of $200\,\mu$m, fitting well within the TEM$_{00}$ cavity mode volume.

The atoms are randomly distributed across the cavity nodes and antinodes, resulting in inhomogeneous single-atom couplings, $g$. At positions of maximal coupling to the cavity, the system is in the regime of small single-atom cooperativity with Purcell factor $C = 4.4 \times 10^{-4} \ll 1$ but large collective cooperativity $NC \gg 1$. A normal mode splitting measurement of $2g\sqrt{N} = 2\pi \times 5.42(14)$ MHz confirms that the collective vacuum Rabi frequency exceeds the relevant decoherence rates in the system: $2g\sqrt{N} \gg \kappa, \gamma$, where $\gamma = 2\pi \times 7.5$ kHz is the natural linewidth of the $^3P_1$, $m_j = 0 \leftrightarrow {}^1S_0$ transition. From the normal mode splitting and atom number, we derive a maximal single-atom coupling,

$g_0 = 2\pi \times 610(30)$, which is used in the simulations (see Supplementary Note 1 and Supplementary Code 1). Averaging the coupling over the standing wave mode of the cavity reduces the collective cooperativity by a factor of 2. Taking into account the finite size of the atomic cloud compared to the cavity waist leads to a further slight reduction of the effective coupling[36].

### Threshold from cavity sub- to superradiance
We investigate SR emission after applying a single pulse of resonant light with varying time duration, $t_P$, corresponding to different excitation angles on the Bloch sphere (see "Methods" section). For this investigation, we reduced the atom number to $N = 20(2) \times 10^6$ to lower the collective Rabi frequency so that SR emission does not start during the excitation pulse. With a $\pi$-pulse duration of 600 ns, we expect that maximally 2% of the atoms decay spontaneously during the excitation pulse. Figure 2a shows a single-shot trace of SR emission after a nominal $3\pi/4$-pulse with peak emitted intensity circled in blue. In Fig. 2b, we plot the normalized average peak emitted intensity (blue) and integral of the pulse (red) of ten traces for various excitation angles on the Bloch sphere. On the x-axis, we plot $\sin^2\left(\frac{\Omega t_P}{2}\right)$, as a proxy for the excited state population at the end of the pump pulse, $\langle \sigma^{22} \rangle_{t=0}$. The maximum population inversion we can achieve is limited by the decoherence in the system (spontaneous emission and Doppler dephasing) during the pump pulse duration. We experimentally determine a nominal $\pi$-pulse as the pump pulse duration which results in the largest peak SR emission.

We observe a threshold excitation angle for a SR burst to be emitted into the cavity mode. Above this threshold, both the peak intensity and pulse area increase approximately linearly, in agreement with the second-order cumulant expansion simulation[37] as shown by the solid lines for the experimental parameters (see Supplementary Note 1 and Supplementary Code 2). If the collective emission rate is limited by the cavity decay rate, $\kappa > 2g\sqrt{N}$, then the peak intensity scales quadratically with positive inversion. However, in our system, the collective emission rate overcomes the cavity decay rate, $2g\sqrt{N} > \kappa$, such that the photons act back on the atoms before escaping the cavity. As a result, the excitations oscillate back and forth between the atoms and cavity before leaving the cavity, as manifest in the slight revival of cavity emission visible in Fig. 2a. In this oscillatory SR regime, the peak scales linearly rather than quadratically with the excited state atom number[17]. Due to imperfections in the excitation pulses such as an inhomogeneous beam profile and intensity fluctuations, measured peak powers near the maximum tend to be biased towards lower values. When targeting excitations below the maximum level, imperfections lead to symmetric deviations in both the peak amplitude and the integrated area of the superradiant emissions. However, in the case of aiming for complete excitation, such as with $\pi$-pulses, any imperfections will result in a reduced population inversion. As a result, measurements up to approximately $\sin^2\left(\frac{\Omega t_P}{2}\right) = 0.9$ exhibit a high degree of agreement, while the agreement gradually levels off beyond this value.

In a system without dephasing, one would expect a threshold at $\Omega t_P = 0.5\pi$, corresponding to $\langle \sigma^{22} \rangle_{t=0} = 50\%$, but due to decoherence in the time before the pulse is emitted, the threshold for superradiance is found to be $\Omega t_P = 0.57\pi$ both in experiment and in simulation, as shown in Fig. 2b. After the SR pulse is emitted, the remaining atoms in the excited state will be subradiant with respect to the cavity mode (in the case of no spontaneous emission this population is equal to $1 - \langle \sigma^{22} \rangle_{t=0}$, as described in[33]). The atoms which are left in the excited state after the SR pulse will then decay spontaneously into free space. If the initial excited state population is below threshold, the atoms are not able to synchronize and their emission into the cavity is suppressed.

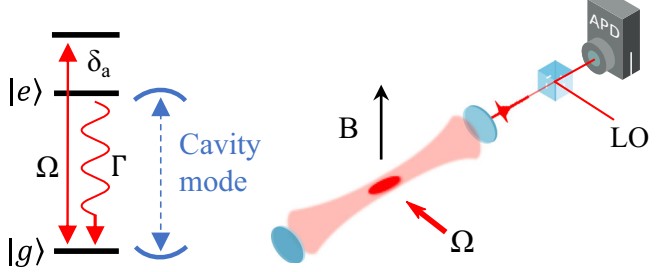

**Fig. 1 | Experiment overview.** Relevant level diagram and experimental schematic. We cool and trap strontium atoms in the center of the cavity. We then pump the atoms at a laser detuning, $\delta_a$, from the $^1S_0 \leftrightarrow {}^3P_1$, $m_j = 0$ 689 nm transition transversely to the cavity axis and observe the intensity of the emitted pulse via a beat detection with a local oscillator.

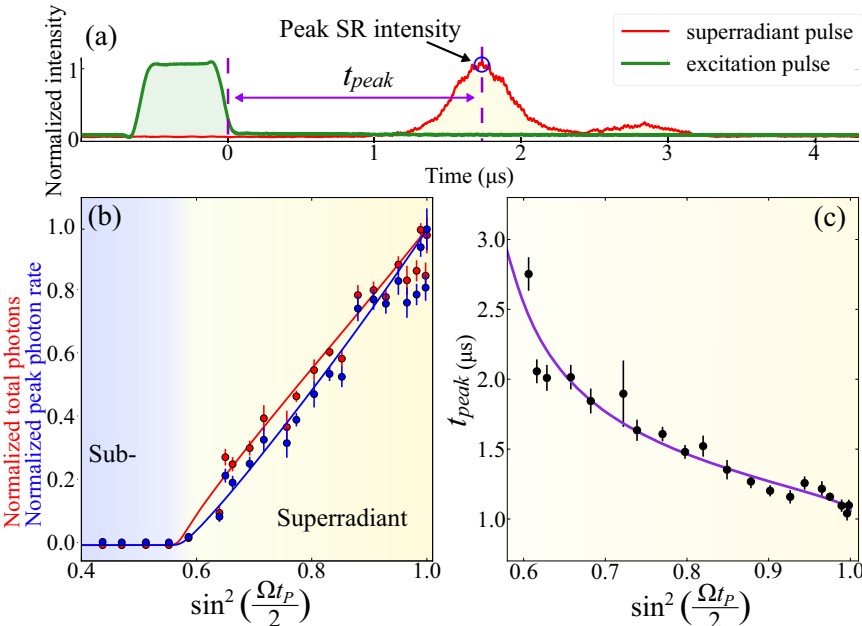

**Fig. 2 | Sub- to superradiant threshold. a** A single-shot trace showing an excitation pulse (green), pulse delay time (purple), and detected SR light pulse (red) along the cavity mode with peak emitted intensity circled in blue. **b** Normalized emitted pulse area (red) and peak amplitude (blue) for varying excitation pulse duration. For excitation pulse angles $\Omega t_P \lesssim 0.57\pi$ no SR pulses are observed, while above this threshold the SR peak amplitude and area scale linearly with pulse angle.

Solid lines are simulations based on the experimental parameters. **c** Delay times of the SR emission for varying pulse angle. The solid purple line is a simulation with a constant temporal offset of 437(17) ns, extending the simulated time delays. Each data point in **b** and **c** is an average of ten measurements with error bars representing the standard deviation of the mean.

Figure 2c shows the delay time between the end of the pump pulse and the peak intensity of the emitted pulse. This is the time required for the individual atomic dipoles to synchronize through the cavity field according to each atom's position in the cavity mode. In agreement with previous research[16], larger inversions result in faster emissions. The purple solid line corresponds to a simulation with a fitted constant temporal offset[15,16] of 437(17) ns, which is added to the simulated delay times. The origin of this offset is most probably simplifying assumptions in the model, such as stationary atomic positions and perfect excitation pulses (see Supplementary Note 1).

If the atoms are excited through the cavity mode, the phases of the atomic dipoles will by necessity match their location in the cavity mode coupling so as to constructively interfere with the cavity mode. In this case, there is no threshold for superradiance and the delay time will be much shorter.

**Collectively enhanced Ramsey readout**

We can exploit the feature that a transverse $\pi/2$-pulse excitation places the sample in a subradiant state, protected from collective decay into the cavity mode. When combined with a SR pulse readout after a second $\pi/2$-pulse, this allows for a cavity-assisted Ramsey scheme as depicted in Fig. 3a. By varying the pump laser detuning, $\delta_a$, and detecting the power emitted from the cavity mode, we plot the average peak amplitude and map out a spectroscopic fringe pattern signal shown in Fig. 3. Each data point is an average of ten repeated measurements, and each measurement is from a separate MOT loading cycle. Figure 3b is a scan around the central Ramsey fringe taken with higher spectral resolution while Fig. 3c is data taken over the complete lineshape.

The blue solid lines are simulations based on the experimental parameters with a rescaled amplitude to account for losses in the beam path. The data show strong agreement with the lineshape predicted by simulation. The width of the envelope of the fringe pattern is given by $\Delta f = 1/\tau_P = 3.33$ MHz, where $\tau_P = 300$ ns is the duration of each $\pi/2$-

pulse. We select a free evolution time $T = 5\mu s$ to balance between being significantly shorter than the natural decay time of 22 $\mu s$ required for a sizable signal, but long enough to observe numerous fringes. The frequency spacing of the Ramsey fringes, or free Ramsey range (FRR), is given by the inverse of the interpulsar free evolution time, $1/T = 200$ kHz. Different from the traditional Ramsey lineshape, the superradiance enhanced lineshape has flat near-zero photon emission zones for detunings which do not result in positive atomic population inversion.

Rabi-type interrogations with durations comparable to the decay rate are limited by SR emission, which begins as soon as positive inversion is achieved. The subradiant behavior of the atoms is essential and ensures protection against cavity decay until after the final $\pi/2$-pulse is applied. It is crucial to excite the atoms perpendicularly to the cavity axis; otherwise, driving them along the cavity mode imparts a relative phase that causes collective radiation into the cavity for any excitation fraction[38].

In a locking scheme, the laser frequency or phase can be incrementally adjusted around a fringe to increase the sensitivity to detuning. By analyzing consecutive measurements, it is possible to generate a feedback signal that guides the laser frequency toward the atomic resonance. Notably, the collectively enhanced lineshape exhibits a distinct kink at the sub- to superradiance transitions, which may serve as a highly precise and narrow feature for laser locking purposes (see Supplementary Note 2). Currently, the observed statistics of the peak amplitude around these kinks show less signal-to-noise (SNR) than around the center of the fringes.

**Nondestructive readout for repeated interrogation**

Each Ramsey sequence causes maximally two recoils per atom, one from the two Ramsey excitation pulses (perpendicular to the cavity axis), and one from the SR emission (along the cavity axis). Without cooling between sequences, we can conduct about ten sequences on the same atomic ensemble. However, each measurement has reduced

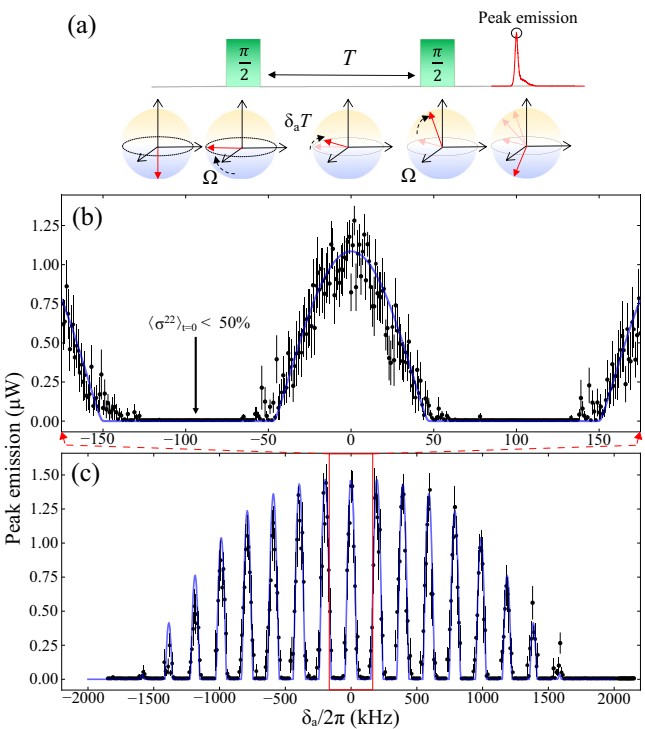

**Fig. 3 | Collectively enhanced Ramsey spectroscopy. a** Diagram of a Ramsey interrogation sequence which uses two $\pi/2$-pulses separated by a free evolution time $T$. Bloch spheres below the sequence show the collective Bloch vector at corresponding times in the sequence which can result in a SR pulse. The first $\pi/2$-pulse pulse brings the atoms to a 50% fractional excitation, at which point the excitation is protected from cavity emission during the free evolution time. A second $\pi/2$-pulse brings the excitation above or below 50% depending on the phase accumulated, $\delta_a T$. If positive inversion is reached, a SR pulse is emitted, shown as a collective Bloch vector accelerated downward on the final Bloch sphere, with peak emission corresponding to the amount of positive inversion. We excite atoms with varying laser detunings and detect SR peak emissions along the cavity axis. **b** A narrow scan around the center fringe, boxed in red in (c), taken with high resolution. **c** Scan taken over the complete spectroscopic lineshape. The blue solid lines are fits from simulation with an overall rescaled amplitude. Each data point is a mean of ten measurements, with separate MOT loading cycles. Error bars indicate standard deviations of the mean.

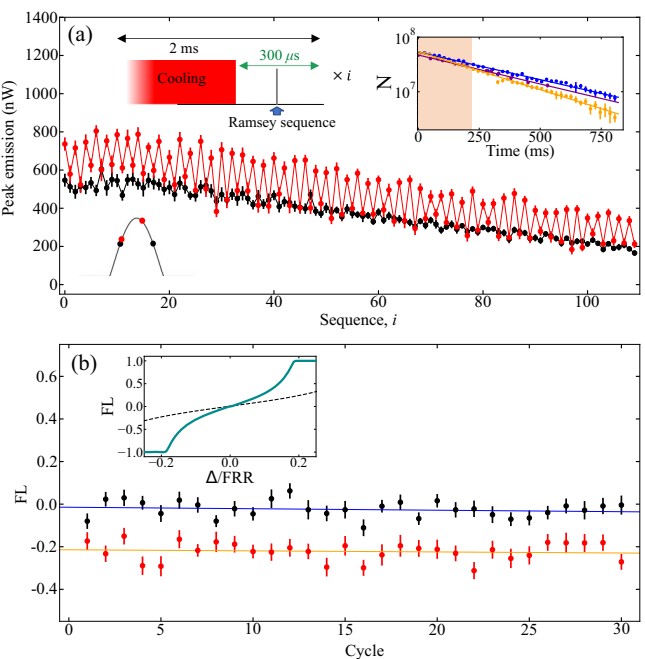

**Fig. 4 | Subsequent Ramsey interrogations for frequency discrimination. a** Peak SR emitted intensities while stepping the interrogation laser frequency between Ramsey sequences in the case of a resonant (black) and detuned laser (red) with error bars representing the standard deviation of the mean of 30 MOT loading cycles. Top left inset: timing diagram for interleaved cooling and Ramsey interrogation. We can perform more than 100 repetitions within a MOT loading cycle. Top right inset: atom number measurements via shadow imaging taken at different times after the start of the repeated Ramsey measurements for the case of blocked Ramsey pump pulses (blue), on-resonant Ramsey pulses (yellow), and for reference, leaving on the MOT the entire time with no pump pulses (purple). Each data point is an average of ten measured atom numbers with error bars representing the standard deviation of the mean. **b** Frequency locator averaged over the 108 pulses in each trace taken in pairs. Error bars are standard deviations of the mean of the 54 pairs of pulses. Solid lines are best-fit lines, consistent with zero slopes, proving no frequency chirp or systematic detuning for 30 loading cycles taken at 6 s intervals. Inset: conversion of FL to laser detuning given our step size and fringe width for the collectively enhanced Ramsey lineshape (cyan) and a traditional Ramsey lineshape (black dotted) for reference.

contrast due to heating, and the resonance shifts each sequence as the atoms are accelerated away from the pump laser direction.

To remove the systematic Doppler shift resulting from consecutive Ramsey sequences, we employ a 2 ms cycling time comprising 1.7 ms of cooling on the single-frequency 689 nm MOT, and 300 $\mu$s for Ramsey interrogation and readout. The timing is illustrated in the top left inset in Fig. 4a. The top right inset shows the atom number estimated from shadow images after a varying number of 2 ms sequences, with Ramsey excitation pulses (yellow) and without (blue), as well as with the 689 nm MOT left on continuously (purple). We observe a decay constant of 313(4) ms when resonant excitation pulses are applied, and 431(9) ms when the excitation pulses are blocked. This indicates a slight reduction in recapture due to heating from the resonant Ramsey sequences. The decay constant of the atoms with the 689 nm MOT left on continuously is 409(9) ms, limited by background gas collisions. Therefore, turning off the MOT for 300 $\mu$s during each sequence does not result in increased losses.

A 2 ms cooling and interrogation sequence is much shorter than the ~1 s time to trap and cool a completely new atomic ensemble, drastically increasing the repetition rate. The Ramsey sequence time is still short compared to the 2 ms, and is limited by the natural lifetime of $^3P_1$. However, this interrogation time can be longer than the cooling

time if we use a more narrow clock transition such as $^3P_0$ and place the atoms in a lattice to counteract gravity.

To demonstrate a preliminary clock feedback measurement, we step the frequency of the interrogation laser between Ramsey sequences within a MOT loading cycle. In Fig. 4a we show SR peak intensities as we toggle the frequency symmetrically around the resonance (black) and around a detuned frequency (red). Averaging over 30 MOT loading cycles, it is evident that in the off-resonant case, the frequency step nearer to atomic resonance consistently yields higher peak emissions. Both sets exhibit a decrease in intensity due to the gradual loss of atoms over time.

We define a frequency locator, $FL = \langle \frac{P_{i+1}-P_i}{P_{i+1}+P_i} \rangle$, where $P_i$ is the peak intensity of the $i$th pulse within a cycle. The FL is the difference between consecutive peaks divided by the sum, to normalize for variation in atom number. We average this value in non-overlapping pairs of consecutive Ramsey sequences over an experimental cycle. Given the free evolution time and frequency stepping size, this value can be converted to an interrogation laser detuning and used for laser frequency feedback. The inset shows how the FL can be related to a detuning when the step size is 0.1 FRR, where the solid line corresponds to the collectively enhanced lineshape and the less steep dotted line is derived from a traditional Ramsey lineshape for

reference. The chosen step size represents a trade-off between maximizing dynamic range and optimizing SNR of our fringes. In a sensor, the ideal step size will be chosen to conform with the requirements and stability of the system (see Supplementary Note 2). In Fig. 4b, we plot the FL averaged over 108 pulses for each MOT loading cycle. Best-fit lines show slopes consistent with zero for both sets, indicating there is no systematic change in frequency throughout the 30 MOT loading cycles taken at 6 s intervals. The off-resonant case has a constant offset, signifying a constant interrogation laser detuning. The 30 traces are taken over the course of three minutes and the deviations in the data points represent frequency excursions of the interrogation laser on this time scale.

The observed readout noise is above the atomic projection noise. How much of this excess noise is due to interrogation laser phase fluctuations and how much stems from inherent stochastic dynamics of SR emission is an important question for applications and is subject to further investigation.

## Discussion

Our study demonstrates the use of collective atomic decay of ultracold Sr for state detection. The collective atomic interaction behaves in agreement with a sub- to superradiant transition with respect to emission in the cavity mode as a function of the ensemble inversion. This behavior can offer a superradiant enhanced readout of a Ramsey interrogation sequence, producing a fast and directional state readout with no additional lasers besides the interrogation laser. This makes the scheme useful for a wide range of sensors.

Other operations such as spin-echoes should be possible without initiating collective cavity decay if these pulses are applied with the appropriate phases such that the collective Bloch vector remains below the equator. Torquing the collective Bloch vector above the equator is permissible if it is done with durations much shorter than the inverse collective Rabi frequency because that is the rate at which atoms will start to superradiantly emit once above threshold. A SR readout scheme presents an active alternative to non-destructive measurement methods, where a single or multiple transitions can be simultaneously interrogated, and results in near-perfect collection efficiency of the emitted photons. This measurement method can be used in optical clock systems or sensors to get fast readouts with less than one scattered photon per atom.

If we apply the technique to an ultranarrow clock transition, such as the $^3P_0$ transition in Sr or Yb, the time spent measuring the atomic state could exceed the necessary cooling times, which could greatly reduce the Dick effect[39]. The higher duty cycle could be particularly beneficial for transportable optical clocks, where the fractional instability is limited by the performance of the clock laser and the associated aliasing of laser noise[40]. Compared to other methods developed to overcome dead-time effects, such as interleaving interrogation of two independent clocks[41], imaging in tweezer arrays[42,43], and non-destructive dispersive probing[9], this method uses no secondary vacuum chamber, high numerical aperture lenses, or extra laser frequencies. It only requires a readout cavity and is compatible with existing cavities used for spin-squeezing[44]. To implement this approach in optical lattice clocks, we propose adding a separate state-readout cavity perpendicular to the necessary 1D lattice confinement axis. One could also implement the method in a shelving scheme, where atoms are shelved on the $^3P_0$ state before a superradiant readout on the $^3P_1$ transition. To be used as an absolute frequency reference, rigorous investigation of systematics is required. However, we do not foresee any significant new types of systematic frequency shifts compared to other traditional optical lattice clocks.

Further investigations will focus on reducing shot-to-shot fluctuations by quantifying the competing emission modes that contribute to the inherent pulse statistics of the SR signal. By combining characteristics such as delay time and peak intensity, a more precise determination of atomic inversion may be obtained. The immediate benefits offered by our scheme could be applied to any quantum sensor relying on the population difference readout of a quantum state. By judicious choice of parameters such as finesse and cooperativity of the readout cavity, our scheme can be implemented in other types of sensors using quantum emitters of varying ensemble sizes.

## Methods

### Transverse excitation

After cooling the atoms, we turn off the MOT laser beams and quadrupole magnetic field, leaving only a vertical bias field from a set of Helmholtz coils. This field of B = 0.2 mT provides a quantization axis and separates each of the $m_j = \pm 1$ Zeeman sublevels of $^3P_1$ by a splitting of $\Delta = 2\pi \times 4.2$ MHz. This together with the choice of polarization direction along the bias field, reduces the dynamics to a simple two-level system. The atoms are driven transversely to the cavity axis, as illustrated in Fig. 1a, with a laser at frequency $\omega_l$, detuning $\delta_a = \omega_l - \omega_a$, and corresponding Rabi frequency $\Omega = 2\pi \times (833 \pm 30)$ kHz. We assume a uniform Rabi frequency since the pump laser waist is much larger than the atomic cloud and it is large compared to the Doppler width of the cold sample.

### Superradiance detection

Emitted power into the cavity mode is detected on an avalanche photodetector via a heterodyne beat measurement with a stable local oscillator (LO) to reject the cavity locking signal.

## Data availability

All data supporting the findings of this study are available from the authors upon request.

## Code availability

The codes that support the findings of this study are available from the authors upon request.

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

## Acknowledgements
We thank Mikkel Tang and Asbjørn A. Jørgensen for their contributions to the experimental apparatus as well as helpful discussions. Figure 1 includes components adapted from ComponentLibrary by Alvise Vianello, available at https://github.com/amv213/ComponentLibrary, and used under a Creative Commons Attribution-NonCommercial 4.0 International License. This project was supported by the European Union's (EU) Horizon 2020 research and innovation program under the Marie Sklodowska-Curie Grant Agreement No. 860579 (MoSaiQC) and Grant Agreement No. 820404 (iqClock project), the USOQS project (17FUN03) under the EMPIR initiative, and the Q-Clocks project under the European Commission's QuantERA initiative. We acknowledge funding from VILLUM FONDEN via Research Grant No. 17558.

## Author contributions
E.A.B., S.L.K., S.A.S., J.R.T., T.Z., J.W.T, and J.H.M. contributed to the overall operations of the experiments. C.H. and H.R. carried out the simulations for the superradiant threshold, time delay, and collectively enhanced Ramsey fringe plots. All authors discussed the results, contributed to the data analysis and worked together on the manuscript.

## Competing interests
The authors declare no competing interests.
