## [Peer Review File · Nature Communications]

Collectively enhanced Ramsey readout by cavity
sub- to superradiant transitionReviewer #1 (Remarks to the Author):

In the manuscript "Collectively enhanced Ramsey readout by cavity sub- to superradiant transition" Bohr et. al. describe experiments on the observation of a minimum excitation threshold of atomic excitation for superradiance to occur in the presence of a pumping field external to the cavity field. They apply this phenomenon to Ramsey spectroscopy of atomic resonances and demonstrate key advantages in atomic state readout with higher sensitivity.

The experiments are carefully carried out and some of the findings are novel: The existence of a threshold for excited state population for superradiant emission into the cavity mode does indeed allow a fundamentally new way to readout atomic states. This has the potential to be applied widely for atomic sensors with added benefits of speed and sensitivity with the added experimental overhead of needing a moderately high finesse cavity. I am inclined to recommend publication in Nature Communications; however I would like to see several important points (in my view) to be addressed and improvements in the presentation for the aid of a reader.

Firstly, I suggest that Figure 1(b) be relocated to a later point in the manuscript, as the results related to the corresponding pulse sequence are elaborated upon only in Figure 3.

The manuscript states a transition from a sub- to superradiant state across a threshold as an existing prediction in the abstract and also in the Introduction with references 27-35, but I am unable to pinpoint where this prediction is distinctly made, as the cited papers are written in very different contexts, presenting many different physical phenomena. Also what is missing is the physical picture of what causes this threshold, which may be obvious to the authors, but not to a reader. Many related phenomena have similar thresholds for superradiance, for instance matter-wave superradiance and while it is somewhat intuitive why superradiant emission requires this threshold, it is not clear to me why the state should be subradiant below this threshold i.e., the emission is suppressed relative to the uncorrelated atom emission. Also how do these states manifest experimentally? For instance in Fig 2(a) a typical above-threshold, superradiant pulse is shown. How does it look like for subradiant and independent-atom emission scenario, even in principle? In my opinion, a deeper look at the subradiant states is warranted. A good schematic of this physical picture can supplement Fig 1.

Fig.2 (a) could show several pulses for a number of atomic excitations, rather than just one. This would give what follows better readability.

I would like to see a more thorough description of the simulations that produce the solid lines in Fig. 2(b) and 2(c). While solid lines generally describe the trends well, there is great deviations between the data and the simulations. Part of this is explained, e.g., close of the peak atomic excitation, but there are also substantial deviations in the blue curve of Fig2(b) between 0.6 -0.8 of x-axis. Same for the Fig.2(c). Is it simply a case of insufficient signal to noise ratio or something else at play?

I find the cooling procedure between Ramsey sequences and its description in the manuscript a distraction. While I appreciate its utility, it is not clear how this is related to the main findings of the field (Figs 2 and 3).

Frequency discrimination comparison between conventional and superradiance-enhanced readouts warrant a more detailed description, since this will be at the heart of any practical sensor based on these. While the new technique gives steeper FL, there are plateau regions beyond ~ 0.2 FRR.

For the application of this technique, some commentary on how this technique is compatible or amenable to quantum sensors of interest is desirable. Can a state of the art optical lattice clock compatible benefit from this without being incompatible? Or a magnetometer? The system described of course takes advantage of long-lived excited states of Sr, but most practical quantum sensors do not use it. What are possibilities of adopting these techniques in those contexts?

In conclusion, I commend the authors for their insightful work and recommend the manuscript's publication. However, I encourage addressing the aforementioned points to enhance the manuscript's clarity and utility for readers.

Reviewer #2 (Remarks to the Author):

Report on 441152_NC, "Collectively enhanced Ramsey readout by cavity sub- to superradiant transition" by E. Bohr and co-authors

The manuscript of E. Bohr and co-authors, entitled "Collectively enhanced Ramsey readout by cavity sub- to superradiant transition" reports on a Ramsey interferometer using an ensemble of ultracold atoms coupled to a cavity. The light-matter coupling mediated by the cavity enhances the amplitude of the signal at the output of the interferometer. Interesting applications of the scheme could be envisioned in metrology and quantum sensing in general. The experimental results are of good quality and convincing. The paper is well-written and comprehensive. As far as I can judge, the references are accurate.

My major concern is about the physical interpretation of the experimental results. The authors interpret the occurrence of emission as a manifestation of superradiance, and the lack or quench of emission as due to subradiance. The main argument to support this claim is a large cooperativity parameter, i.e. $CN \gg 1$. On the other hand, one might also interpret the results as a lasing effect which kicks in when the population inversion is achieved. This alternative interpretation can be also supported by the linear dependence of the emission with the atoms number. One can regret that no systematic studies as function of the atom number are presented to reinforce the author's interpretation. The authors should address this issue, and ideally comfort their analysis with new data, before considering the manuscript for publication in Nat. Comm.

In addition to my main concern, I have other minor remarks which are classified as their appear in the manuscript

(1) In Introduction section, the authors claim that "the subradiant behavior allows for long interaction times necessary for resolving and exploiting ultranarrow clock transitions." It does not appear very clear, what is the expected improvement in open cavity environment.

(2) In Section 3, the authors wrote "... in agreement with the simulation as shown by the solid lines for the experimental parameters." For the shake of self-consistency, would be interesting to have more details about the simulations and the model. Which parameters are set by the experiment and which ones are left free.

(3) In Section 3, the authors wrote "Due to imperfections in the n -pulse excitation...". What kind of imperfections the authors have in mind, and why they are visible only close to π -pulse?

(4) In Section 3, the authors wrote "...leaving them in a subradiant state". My question here is connected to my major concern. What is the experimental signature showing that the system is in a subradiant state? My intuition is that in such open system, the decay rate is dominated by single atom decay.

(5) In Section 3, the authors wrote "The disagreement is likely due to simplifying assumptions in the model." Can the authors be more specific on which simplifications lead to such a disagreement?

(6) In Section 4, the authors wrote "...with a rescaled amplitude." Why such a rescaling is necessary and what is its magnitude? Does it question the agreement with the model?

(7) In Section 4, the authors wrote "The subradiant behaviour of the atoms is essential and ensures protection against cavity decay until after the final $\pi/2$ -pulse is applied. For Ramsey excitation through the cavity, the protection via subradiant states is lacking due to the inherent phase-matching, significantly reducing the possible free evolution time." It appears to me that the two sentences are in contradictions. Can the authors clarify this point?

(8) For frequency measurements, it is of crucial importance to understand and measure systematic shifts. Would be interesting to have (a brief) discussion on this issue in the conclusion. For instance, what are the origin of the systematics; Cavity pulling, atom number, delay time, ...?

Reviewer #3 (Remarks to the Author):

Summary

Bohr et al present a cavity superradiance based atomic state detection method and discuss its applications to quantum sensors such as atomic clocks. By placing millions of neutral bosonic strontium atoms inside a moderate finesse cavity, the authors engineer a cavity QED system in the strong collective cooperativity regime. With sufficient population inversion, the atoms emit a superradiant burst of light into the cavity mode while decaying to the ground state. Since this requires more than 50% population in the excited state, the cavity does not decohere atoms during the preparation and interrogation time of a Ramsey-based quantum sensor, and so can be used to enhance final state readout. The authors demonstrate this readout scheme and generate a laser-frequency-dependent error signal with which they could, in principle, lock the laser to the atomic transition frequency.

The physics associated with the superradiant readout scheme has been previously observed and that work is cited in the manuscript. This review is therefore more focused on its applications to atomic readout in metrology. The authors provide a valuable insight in articulating the application of cavity superradiance to readout in Ramsey-type sequences. This scope is appropriate to the journal. However, I found various aspects of the discussion supporting the stated claims to be lacking (see general comments below). Therefore, I cannot recommend publication until significant improvements are made to the framing, contextualization, and discussion of the results.

General comments

- (1) In clearly defined quantitative terms, what are the advantages and disadvantages of this readout scheme?
- (2) How does this readout scheme compare to straightforward atomic state detection in free space (it is asserted in the abstract that this method has "high-sensitivity" and later, in the introduction, that the cavity "drastically increases detection efficiency compared to free-space" fluorescence detection, but this claim is not clearly substantiated in the body of the manuscript)? The frequency discrimination implied by the error bars in figures 3a and 4 does not appear especially precise, and in any event should be compared quantitatively to standard readout with the same resources (e.g. atom number) in order to substantiate a claim of increased efficiency and/or sensitivity.
- (3) Instead, my understanding of the key result is that the scheme enables repeated highly non-destructive readout in less than two milliseconds, dominated by the 1.7 ms of cooling. Indeed non-destructive readout in free space often takes at least 10 ms or longer. Furthermore, the actual detection time is only a few microseconds. Given the relatively short excited state lifetime used in this work, this is a key enabling advantage. Yet this context is not clearly presented.
- (4) Some discussion of the limitations of this scheme should occur. For instance, more complicated sensing schemes involving Hahn echos or various dynamical decoupling sequences may not be possible since the atomic spin may become too inverted. Also, the inversion region for phase estimation is reduced in this readout scheme since all population fractions below threshold yield zero signal.

Detailed comments

- (5) Various quantities are vague and should be stated precisely, with error bars.
 - a. Cavity decay rate of 780 kHz lacks an error bar.
 - b. Why is a firm number for the finesse not provided? Cavity linewidth and free spectral range are both easily measured.
 - c. Stated atom numbers presumably have a significant error bar which should be included.
 - d. Normal mode splitting is said to be "highly resolved" but the data supporting this statement are not

shown. At the very least the fitted $\sqrt{N} \cdot g$ with error bar should be provided. This data is later referenced to provide an expectation for the single-atom-single-photon coupling g . This further emphasizes the need for error bars! The simulated g and fitted g may or may not be in conflict with each other if proper error analysis is performed.

(6) Imperfect π -pulses are referenced as the primary limitation for the maximum inversion and therefore the maximum detected signal.

a. How much of a reduction do you see? How much of that can be attributed to excited state decay during the excitation time?

b. From the statement "Slight deviations from a π -pulse always result in lower peak emitted intensity" I infer that your model is temporal intensity noise causing the Rabi frequency to fluctuate between pulses. If so, this is a slow variation which can be measured – is the observed reduction in π -pulse contrast consistent with these intensity fluctuations?

c. Or perhaps the model is different. If this is the case, what is the model and is the observed π -pulse fidelity consistent with experimental sources of those errors.

(7) The authors have used a simulation to provide a theory curve in figures 2 and 3 and to provide comparison numbers in the main text. This simulation is not described anywhere. The contents and details of this model must be stated (what are the Hamiltonian and Lindblad terms, what parameters are fixed by which calibration, what parameters are left as free parameters to be fit, etc.).

(8) I am confused by the sentence "Immediately after the SR pulse is emitted, the atoms retain an excited state population of ...". This sentence does not appear to serve a purpose. Also, as defined earlier, $\langle \sigma_{22} \rangle_{t=0}$ is the excited state population at the end of the pump pulse. As mentioned earlier in the paragraph, there is some decay between the end of the pump pulse and the superradiant pulse so I would expect the excited state population after this decay to play a role rather than the original excited state population. But, regardless, I would further expect the excited state fraction after the superradiant pulse to be simply zero. **except that atoms in ground state can absorb some of the superradiant pulse"

(9) The sentences beginning with and following, "Notably, the collectively enhanced lineshape exhibits a distinct kink..." argues that the threshold for superradiant emission would provide a good narrow error signal for locking a laser to atoms. This is a surprising argument since the emission is uniformly zero for all phase deviations below this threshold, so this error signal would provide no information as to whether the laser is exactly at the lock point or significantly off in one direction. Indeed the error signal proposed, the frequency locator, operates much more conventionally where the signal is large and with a high sensitivity to small changes in the laser phase.

(10) The sentence "Also, it is possible to get 100s of useful pulses..." is colloquial and vague. What is meant by useful? The data in figure 4a appears to show almost a factor of three reduction in atom number after 100 (not 100s) repeated pulses. It is not clear what atom number I should compare to.

(11) How was the frequency locator detuning of 0.1 FRR chosen? Is this optimized to provide the largest derivative at zero laser detuning?

(12) In the outlook, it is suggested that the readout scheme could be applied to the clock transition in Yb or Sr, which could yield advantages in reducing Dick effect noise. I agree that this could be valuable, especially for many-times-repeated short interrogation time clocks. It is worth noting that there has been work to minimize Dick effect noise in optical lattice clocks (e.g. <https://arxiv.org/abs/1305.5869>) and tweezer clocks (e.g. <https://arxiv.org/pdf/1904.10934.pdf> and <https://arxiv.org/pdf/1811.06014.pdf>). It might be useful to include some comparative statement with alternative methods to reduce this source of noise.

REVIEWER COMMENTS

Reviewer #1 (Remarks to the Author):

In the manuscript “Collectively enhanced Ramsey readout by cavity sub- to superradiant transition” Bohr et. al. describe experiments on the observation of a minimum excitation threshold of atomic excitation for superradiance to occur in the presence of a pumping field external to the cavity field. They apply this phenomenon to Ramsey spectroscopy of atomic resonances and demonstrate key advantages in atomic state readout with higher sensitivity.

The experiments are carefully carried out and some of the findings are novel: The existence of a threshold for excited state population for superradiant emission into the cavity mode does indeed allow a fundamentally new way to readout atomic states. This has the potential to be applied widely for atomic sensors with added benefits of speed and sensitivity with the added experimental overhead of needing a moderately high finesse cavity. I am inclined to recommend publication in Nature Communications; however I would like to see several important points (in my view) to be addressed and improvements in the presentation for the aid of a reader.

We thank the Referee for the insightful feedback and recognition of the novelty and potential impact of our findings. We implemented several changes to enhance the presentation, as detailed below.

Firstly, I suggest that Figure 1(b) be relocated to a later point in the manuscript, as the results related to the corresponding pulse sequence are elaborated upon only in Figure 3.

We agree with the Referee's suggestion regarding the placement of Figure 1(b) and have relocated Fig. 1(b) to Fig. 3, and adjusted the captions and references to the figures in the text accordingly. We believe this makes for a more timely appearance in the manuscript and makes for a clearer standalone understanding of Fig. 3.

The manuscript states a transition from a sub- to superradiant state across a threshold as an existing prediction in the abstract and also in the Introduction with references 27-35, but I am unable to pinpoint where this prediction is distinctly made, as the cited papers are written in very different contexts, presenting many different physical phenomena. Also what is missing is the physical picture of what causes this threshold, which may be obvious to the authors, but not to a reader. Many related phenomena have similar thresholds for superradiance, for instance matter-wave superradiance and while it is somewhat intuitive why superradiant emission requires this threshold, it is not clear to me why the state should be subradiant below this threshold i.e., the emission is suppressed relative to the uncorrelated atom emission. Also how do these states manifest experimentally? For instance in Fig 2(a) a typical above-threshold, superradiant pulse is shown. How does it look like for subradiant and independent-atom emission scenario, even in principle? In my

opinion, a deeper look at the subradiant states is warranted. A good schematic of this physical picture can supplement Fig 1.

The prediction of the sub- to superradiant threshold for a transversely excited ensemble inside a cavity is made in Ref. [21]. References [27-35] are included to highlight experimental observations of subradiance in varied systems and platforms, such as free space atoms, quantum dots, and molecules, and to illustrate the breadth and applicability of the effect.

For a detailed physical illustration, we point to Fig. 3(a) and Fig. 4 in Ref. [21]. The predicted subradiant behavior is subradiant only relative to the cavity mode, and the atoms still undergo spontaneous decay into free space. We agree that a direct measurement of this subradiant behavior would indeed be revealing and introduce captivating physics. However, we calculate for our atom number that spontaneous emission on the kHz transition into the solid angle of the cavity mirrors would produce an intensity of ~ 100 pW (given our low cooperativity). Directly measuring subradiance would require distinguishing powers in the 10s of pW range, which is beyond our current detection capabilities. For this work, the important aspect is that we can accumulate a phase difference between two Ramsey pulses without collective cavity decay.

While we do not assert that we measure subradiant powers, our observations are in accord with the expected behavior detailed in Ref. [21]. To facilitate a better understanding for the readers, we have now revised and included additional sentences in the introduction to clarify this point and make explicit reference to the theory paper at the end of page 2: *“Below this threshold, the atoms are predicted to exhibit suppressed emission, or subradiance [27–35], with respect to the cavity mode [21]. Emission into free space remains unchanged, and the atoms are still susceptible to the single-atom spontaneous decay into modes outside of the cavity’s solid angle.”*

Fig.2 (a) could show several pulses for a number of atomic excitations, rather than just one. This would give what follows better readability.

We appreciate the reviewer's suggestion regarding Fig.2(a). Our primary intent in this figure was to clearly illustrate specific attributes of the pulse, such as peak intensity and the time delay before the peak pulse intensity. While showing multiple pulses might provide insight into the pulse statistics, it could also potentially obscure the individual pulse characteristics that we aimed to emphasize. Therefore, we decided to keep Fig. 2(a) as it is.

I would like to see a more thorough description of the simulations that produce the solid lines in Fig. 2(b) and 2(c). While solid lines generally describe the trends well, there is great deviations between the data and the simulations. Part of this is explained, e.g., close of the peak atomic excitation, but there are also substantial deviations in the blue curve of Fig2(b)

between 0.6 -0.8 of x-axis. Same for the Fig.2(c). Is it simply a case of insufficient signal to noise ratio or something else at play?

We have added supplementary material describing the simulations and model in detail, and have added a reference to this in the text on page 4.

In response to the Referees' feedback about the comparison between data and simulation in Fig. 2, we have made some adjustments. Rather than inferring the single-atom coupling, g , from the time delay data, we now employ our directly measured g from our normal mode splitting measurement. This provides a more reliable representation of our system's parameters. For the time delay plot in Fig. 2(c) we introduce a constant time delay offset [15,16]. The origin of this offset is most probably simplifying assumptions in the model, such as stationary atomic positions and perfect excitation pulses.

Addressing the specific deviations noted between 0.6-0.8 on the x-axis: with the updated value of g , the simulation agrees now much better with these data points. While our model presumes the atoms are at absolute zero temperature and lack any transverse width distribution across the cavity mode, we believe that the primary objective, which is to elucidate the general trend of a linearly increasing pulse amplitude suitable for atomic state readout, remains clear and intact.

I find the cooling procedure between Ramsey sequences and its description in the manuscript a distraction. While I appreciate its utility, it is not clear how this is related to the main findings of the field (Figs 2 and 3).

Our intent in the experimental realization of the cooling procedure between Ramsey sequences was to highlight the speed and minimal heating introduced by the novel atomic state readout scheme. We believe this is one of the key advantages of this scheme. This is in contrast to a common fluorescence readout scheme such as electron shelving for neutral atoms, where the large number of scattered photons would typically render an additional interrogation of the atomic ensemble impossible.

Frequency discrimination comparison between conventional and superradiance-enhanced readouts warrant a more detailed description, since this will be at the heart of any practical sensor based on these. While the new technique gives steeper FL, there are plateau regions beyond ~ 0.2 FRR.

We thank the Referee for emphasizing the importance of describing the new frequency discriminator (FL) lineshape. There are indeed plateaus in the discriminator beyond ~ 0.2 FRR. It is crucial to step the frequency such that there is always at least one superradiant pulse in an FL measurement. Optimizing the technique involves choosing frequency steps to obtain a suitable balance between the FL slope and the signal-to-noise ratio (SNR) to form a deterministic measurement of the frequency deviation. With an improved SNR, the frequency

steps can be tuned to obtain the steepest slope by probing close to the kinks. A realistic possibility would be to incorporate a series of measurements with different step sizes, or different free evolution periods, T .

Below in Fig. 1, we simulate the discriminator shape for various stepping sizes ranging from 0.01 FRR to 0.5 FRR. For a step size of 0.5 FRR, which corresponds to interrogating at the kinks, there is a very steep slope but with very little dynamic range. As we decrease the step size the slope decreases and dynamic range increases. As such the step size must be chosen to conform with the requirements and stability of the system in question. We have decided to include a supplemental material note (Supplementary Information, Section 2) including this figure to give the reader a better understanding of the influence of step size on the FL. The new lineshape warrants further investigation of optimal phase/frequency stepping and increasing the SNR of the spectroscopic lineshape before qualified judgment can be made, which is the current subject of investigation in our lab.

Figure 1: Conversion slope for various frequency step sizes.

In addition to adding supplemental material, to address this in the paper we have added the following sentence on page 7: “*The chosen step size represents a trade-off between maximizing dynamic range and optimizing SNR of our fringes. In a sensor, the ideal step size will be chosen to conform with the requirements and stability of the system (Supplementary Information, Section 2).*”

For the application of this technique, some commentary on how this technique is compatible or amenable to quantum sensors of interest is desirable. Can a state of the art optical lattice clock compatible benefit from this without being incompatible? Or a magnetometer? The system described of course takes advantage of long-lived excited states of Sr, but most practical quantum sensors do not use it. What are possibilities of adopting these techniques in those contexts?

The superradiant Ramsey readout scheme is a detection technique that can only be realized in systems where a superradiant emission is possible. This requires that the system in question can fulfill the requirement for the collective decay rate to be much larger than any decoherence in the system. As such, the method is well suited for large ensembles of atoms, or ensembles with a large cavity interaction strength. Energy levels with long lifetimes compared to the collective emission rate limit spontaneous emission into the environment, producing a high-quality signal in the cavity mode. This can in principle be achieved for any lifetime, but is particularly practical for metastable states.

In an atomic clock, the superradiant emission can sometimes be observed and is treated as a loss of signal. It can be forced by adding an optical cavity of sufficient Q-factor, such as those more and more commonly used for other state detection methods or in order to achieve power buildup of an optical lattice. With our approach that decay becomes an asset, and is compatible with cavities used for spin-squeezing in clocks (<https://arxiv.org/abs/2211.08621>). We have added in this reference in the conclusion: *“It only requires a readout cavity and is compatible with existing cavities used for spin-squeezing [43].”*

Superradiance is particularly useful for high-precision magnetometers because it is possible to realize superradiant emission on more than one transition at a time. Rather than alternating between two stretched magnetic states, a superradiant detection method can generate superradiant emission on both lines simultaneously, detectable by monitoring a beat frequency in the output light.

Because the method relies on inversion, it is inherently a differential measurement between two states. Therefore one could imagine more advanced quantum sensing devices where a differential measurement can ensure common mode cancellation of noise compared to independent population measurements of the two states.

To include these considerations we have added the following sentence in the conclusion of the manuscript on page 12: *“A superradiant readout scheme presents an active alternative to non-destructive measurement methods, where a single or multiple transitions can be simultaneously interrogated, and results in near-perfect collection efficiency of the emitted photons. The measurement method can be used in optical clock systems or sensors to get fast readouts with less than 1 scattered photon per atom.”*

In conclusion, I commend the authors for their insightful work and recommend the manuscript's publication. However, I encourage addressing the aforementioned points to enhance the manuscript's clarity and utility for readers.

We thank the Referee for recommending the manuscript's publication and we believe that addressing the points raised has strengthened the study significantly.

Reviewer #2 (Remarks to the Author):

Report on 441152_NC, "Collectively enhanced Ramsey readout by cavity sub- to superradiant transition" by E. Bohr and co-authors

The manuscript of E. Bohr and co-authors, entitled "Collectively enhanced Ramsey readout by cavity sub- to superradiant transition" reports on a Ramsey interferometer using an ensemble of ultracold atoms coupled to a cavity. The light-matter coupling mediated by the cavity enhances the amplitude of the signal at the output of the interferometer. Interesting applications of the scheme could be envisioned in metrology and quantum sensing in general. The experimental results are of good quality and convincing. The paper is well-written and comprehensive. As far as I can judge, the references are accurate. My major concern is about the physical interpretation of the experimental results. The authors interpret the occurrence of emission as a manifestation of superradiance, and the lack or quench of emission as due to subradiance. The main argument to support this claim is a large cooperativity parameter, i.e. $CN \gg 1$. On the other hand, one might also interpret the results as a lasing effect which kicks in when the population inversion is achieved. This alternative interpretation can be also supported by the linear dependence of the emission with the atoms number. One can regret that no systematic studies as function of the atom number are presented to reinforce the author's interpretation. The authors should address this issue, and ideally comfort their analysis with new data, before considering the manuscript for publication in Nat. Comm.

We thank the Referee for recognizing the quality and potential applications of our work. The Referee brings up a good question that we need to clarify for the reader in regard to the physical interpretation of our pulses as superradiance. A characteristic sign of superradiance is that the amplitude of the pulse has a quadratic dependence on atom number. For cavity superradiance in the bad-cavity regime, however, this dependence depends on the ratio between the collective vacuum Rabi frequency and the cavity decay rate, as elaborated in our PRA, Ref.[17] (Figs. 3 and 4). In the regime where the cavity decay rate is larger than the collective vacuum Rabi frequency, the amplitude of the pulses scales as N^2 . However, in the regime where the collective Rabi frequency is larger than the cavity decay rate, photons emitted by the atoms get reflected and re-excite atoms before leaving the cavity. This results in ringing of the emitted intensity in the pulses and a peak amplitude proportional to N . Due to our large atom numbers (a few 10^6 atoms), we find ourselves in this linear regime. The same realization and explanation are expressed in (Gogyan et. al, Optics Express Vol. 28, 5, pp. 6881-6892 (2020)).

When generating laser pulses in the superradiant parameter regime ($CN \gg 1$) the N^2 scaling of the peak intensity of the pulse is present for low collective Rabi rates, but becomes linear as the rate approaches the cavity linewidth. Using the term superradiance for these pulses is consistent with prior work in the field, see e.g., Refs.[14] and [16].

To further clarify why this linear behavior is expected in our system rather than a quadratic scaling, we have revised Section 3 on page 4: *“If the collective emission rate is limited by the cavity decay rate, $\kappa > 2g\sqrt{N}$, then the peak intensity scales quadratically with positive inversion. However, in our system, the collective emission rate overcomes the cavity decay rate, $2g\sqrt{N} > \kappa$, such that the photons act back on the atoms before escaping the cavity.”*

In addition to my main concern, I have other minor remarks which are classified as their appear in the manuscript

(1) In Introduction section, the authors claim that “the subradiant behavior allows for long interaction times necessary for resolving and exploiting ultranarrow clock transitions.” It does not appear very clear, what is the expected improvement in open cavity environment.

We acknowledge the potential for ambiguity in our initial phrasing. To clarify, the subradiant behavior we refer to pertains specifically to the cavity mode and is not indicative of any suppressed emission into modes outside of the cavity solid angle. In our study, the atoms exhibit behavior in agreement with subradiance with respect to collective cavity decay. This subradiance does not suppress spontaneous emission into free space but does protect the atoms from the fast collective decay into the cavity mode. For example, our free evolution periods extend to 5 μs , while the superradiant decay typically occurs in about 1 μs . We have revised this sentence in the manuscript to convey this more explicitly on page 3: *“the subradiant behavior with respect to the cavity mode allows for interaction times without collective cavity decay, necessary for resolving and exploiting ultranarrow clock transitions.”* To further clarify this point we have also included a sentence earlier on page 3: *“Emission into free-space remains unchanged, and the atoms are still susceptible to single-body spontaneous decay into modes outside of the cavity's solid angle.”*

(2) In Section 3, the authors wrote “... in agreement with the simulation as shown by the solid lines for the experimental parameters.” For the sake of self-consistency, would be interesting to have more details about the simulations and the model. Which parameters are set by the experiment and which ones are left free.

We have added supplementary material (Supplementary Information, Section 1) which includes a detailed description of the simulation with parameters used along with two programs: one to produce the simulation data shown in the plots, and one to show the equations used.

(3) In Section 3, the authors wrote “Due to imperfections in the π -pulse excitation...”. What kind of imperfections the authors have in mind, and why they are visible only close to pi-pulse?

There are two primary factors contributing to the described imperfections. Firstly, the pumping laser beam passes through a window that has uneven transmittance caused by built-up strontium coating. This results in a non-uniform pump intensity profile which causes spatially varying Rabi frequencies across the atomic ensemble. The bandwidth of the laser locking feedback results in laser phase noise at ~ 1 MHz which coincides with the timescales of our excitation pulses (~ 1 μ s) and significantly contributes to errors in the nominal pi pulse angle.

The implications of these imperfections vary depending on our excitation target. For a 50% excitation fraction, these imperfections result in symmetrically distributed deviations in both peak amplitude and area, which will average out over consecutive measurements. However, when aiming for complete excitation (100%), any deviation manifests as a lower population inversion. This is why the asymmetrical deviations become particularly noticeable close to the pi-pulse. We are optimistic that refining our pump pulses, in terms of beam shape and laser noise mitigation, will help minimize this discrepancy near pi-pulse excitation.

Also, please note that in response to the Referees' comments, we have rerun the simulations using the single-atom coupling from our normal mode splitting measurement, as opposed to a fit. This has led to a stronger overall agreement between data and simulations. However, near the pi-pulse, there is still some deviation.

To make the above point clearer we have included a more thorough explanation on page 4: *“Due to imperfections in the excitation pulses such as an inhomogeneous beam profile and intensity fluctuations, measured peak powers near the maximum tend to be biased towards lower values. When targeting excitations below the maximum level, imperfections lead to symmetric deviations in both the peak amplitude and the integrated area of the superradiant emissions. However, in the case of aiming for complete excitation, such as with π -pulses, any imperfections will result in a reduced population inversion.”*

(4) In Section 3, the authors wrote “...leaving them in a subradiant state”. My question here is connected to my major concern. What is the experimental signature showing that the system is in a subradiant state? My intuition is that in such open system, the decay rate is dominated by single atom decay.

Indeed, the atoms are always susceptible to single-atom decay into free space. When we refer to the "subradiant state", it is specifically in regard to emission into the cavity mode, not spontaneous emission into free space. We have not made a direct observation of

subradiance, but our observations of Ramsey inter-pulsar evolution times without collective cavity decay are in agreement with the expectations for subradiant behavior with respect to the cavity as presented in Ref. [21].

We have added additional sentences to make this clearer in Section 1, page 3: “*Below this threshold, the atoms are predicted to exhibit suppressed emission, or subradiance [27–35], with respect to the cavity mode [21]. Emission into free-space modes outside of the cavity's solid angle remains unchanged, and the atoms are still susceptible to single-atom spontaneous decay.*” We have also added “*with respect to the cavity mode*” in the referred-to sentence in Section 3, page 4, to keep this distinction clear throughout the manuscript.

(5) In Section 3, the authors wrote “The disagreement is likely due to simplifying assumptions in the model.” Can the authors be more specific on which simplifications lead to such a disagreement?

As mentioned in the response above, we have adjusted the simulations to use the single-atom coupling based on our normal mode splitting measurement, which is a much more robust method. To fit the time delay data, we introduce a constant time delay offset [15,16] of 437(17) ns. The origin of this offset is most probably simplifying assumptions in the model, such as stationary atomic positions and perfect excitation pulses.

We have added supplemental material detailing the assumptions and parameters of the simulation. The simulations have been adjusted accordingly and we have added the text in Section 3, page 5: “*The purple solid line corresponds to a simulation with a constant temporal offset [15, 16] of 437(17) ns, which is added to the simulated delay times. The origin of this offset is most probably simplifying assumptions in the model, such as stationary atomic positions and perfect excitation pulses (see Supplementary Information, Section 1).*”

(6) In Section 4, the authors wrote “...with a rescaled amplitude.” Why such a rescaling is necessary and what is its magnitude? Does it question the agreement with the model?

The rescaling accounts for losses in vacuum chamber windows, optics, and detector efficiencies, and does not qualitatively change or lead to a disagreement with the model. Instead of attempting to individually account for each efficiency, we chose to employ a rescaling approach. To improve clarity, we have adjusted the referred-to sentence on page 5: “*The blue solid lines are simulations based on the experimental parameters with a rescaled amplitude to account for losses in the beam path.*”

(7) In Section 4, the authors wrote “The subradiant behaviour of the atoms is essential and ensures protection against cavity decay until after the final $\pi/2$ -pulse is applied. For

Ramsey excitation through the cavity, the protection via subradiant states is lacking due to the inherent phase-matching, significantly reducing the possible free evolution time.” It appears to me that the two sentences are in contradictions. Can the authors clarify this point?

We thank the Referee for pointing out the need for clarity. In the first sentence, we reference the situation where the atoms are excited transversely to the cavity axis, resulting in subradiant behavior that provides protection against cavity decay. On the other hand, the second sentence pertains to the scenario where the atoms are excited through the cavity mode. In the second sentence, we explain that if we excite the atoms through one of the cavity mirrors, then the atoms would be excited with the exact phase pattern to result in superradiant emission for any excitation fraction.

We have revised the second sentence and included a reference to where excitation is done through the cavity mode. The second sentence now reads: *“It is crucial to excite the atoms perpendicularly to the cavity axis; otherwise, driving them along the cavity mode imparts a relative phase that causes collective radiation into the cavity for any excitation fraction [37]”*

(8) For frequency measurements, it is of crucial importance to understand and measure systematic shifts. Would be interesting to have (a brief) discussion on this issue in the conclusion. For instance, what are the origin of the systematics; Cavity pulling, atom number, delay time, ...?

We appreciate the Referee pointing out the importance of understanding and measuring systematic shifts, particularly in the context of frequency measurements. Regarding cavity pulling, our focus in this work is not on the frequency of the emitted light. We infer that the pulling to which the Referee refers pertains to the location of the center fringe in the spectroscopic signal. We have taken data for pulses with a detuned cavity, and the effect on the pulse amplitude only becomes appreciable for detunings of approximately 500 kHz, whereas the cavity is typically locked to within 10 kHz of atomic resonance.

By applying our pump perpendicular to the cavity axis and cooling between readouts, we have mitigated the Doppler shift of our spectroscopic measurement when running multiple sequences per MOT cycle. An offset in the excitation angle would cause a broadening rather than a shift due to the nature of a standing wave cavity mode.

With our high atom densities, $\sim 1e13/(cm^3)$, there is likely some shift of the transition frequency. Density shifts in Sr88 approaching this density were investigated by (Ito et. al, PRL 94, 153001 (2005)) who found the center line shift to be in the 1 kHz per $1e12/(cm^3)$ range.

We believe the delay time for the superradiant peak would not contribute a systematic effect on the center line but is rather an attribute related to the excited state fraction and atom-cavity coupling.

To be used in a sensor or clock, a rigorous budget of systematic frequency shifts is necessary. We anticipate that future work, with enhanced precision, will delve into these systematic issues in detail. We therefore add the following sentences in the conclusion on page 12: *“To be used as an absolute frequency reference, rigorous investigation of systematic effects is required. However, we do not foresee any significant new types of systematic frequency shifts compared to traditional optical lattice clocks.”*

Reviewer #3 (Remarks to the Author):

Summary

Bohr et al present a cavity superradiance based atomic state detection method and discuss its applications to quantum sensors such as atomic clocks. By placing millions of neutral bosonic strontium atoms inside a moderate finesse cavity, the authors engineer a cavity QED system in the strong collective cooperativity regime. With sufficient population inversion, the atoms emit a superradiant burst of light into the cavity mode while decaying to the ground state. Since this requires more than 50% population in the excited state, the cavity does not decohere atoms during the preparation and interrogation time of a Ramsey-based quantum sensor, and so can be used to enhance final state readout. The authors demonstrate this readout scheme and generate a laser-frequency-dependent error signal with which they could, in principle, lock the laser to the atomic transition frequency.

The physics associated with the superradiant readout scheme has been previously observed and that work is cited in the manuscript. This review is therefore more focused on its applications to atomic readout in metrology. The authors provide a valuable insight in articulating the application of cavity superradiance to readout in Ramsey-type sequences. This scope is appropriate to the journal. However, I found various aspects of the discussion supporting the stated claims to be lacking (see general comments below). Therefore, I cannot recommend publication until significant improvements are made to the framing, contextualization, and discussion of the results.

We thank the Referee for the comprehensive review and for recognizing the potential of our application of cavity superradiance in metrology. While the physics was indeed theoretically proposed in Ref. [21], our present work is the first to map out an inversion threshold for superradiant emission for a transversely driven ensemble. This physics, including the threshold and new lineshape, are some of the key findings in addition to the application to atomic readout in metrology. Below are detailed responses to each point from the Referee.

General comments

(1) In clearly defined quantitative terms, what are the advantages and disadvantages of this readout scheme?

Compared to optical cycling after electron shelving which heats up the atoms beyond recapture, our scheme results in at most two photon recoils, along well-defined directions, per interrogation cycle. This allows for the possibility of multiple readouts for each full experimental cycle, especially if the recoils are compensated by cooling. It is also possible to alternately interrogate from opposing sides in order to minimize the necessary cooling.

The scheme offers directional emission of signal photons. Rather than fluorescing in a random direction, there is a strongly preferred decay channel into the cavity mode. For our atom number and cavity parameters, there is a $\sim 10,000$ -fold enhancement of emission into the solid angle of the cavity mirrors compared to random emission. Furthermore, the natural lifetime of the $3P_1$ state is $21 \mu\text{s}$, whereas we gather the photons from this state much faster, within a couple of μs . For narrow transitions, this technique allows the photons to be collected up to 3 orders of magnitude faster, see Ref [15], than with spontaneous decay, and with close to perfect efficiency.

Other techniques that allow reusing the atomic ensemble require additional lasers for atomic population readout. Our method is simple in that it uses no extra lasers, only the photons from the clock interrogation laser.

Two other notable non-destructive methods are “noise-immune cavity-assisted non-destructive detection” (G. Vallet et al., *New J. Phys.* 19 083002 (2017)) which relies on measuring phase shifts via a third atomic state, and “normal mode splitting”-enabled measurements (J. Bohnet et al., *Nature Photonics* 8, 731 (2014)) that rely on ground-state atomic populations. Both methods require an optical cavity.

The first approach allows for measurements within a predetermined range of atom numbers, and uses a third energy level to measure the ground state population. The second scheme requires similar conditions to the scheme we propose here: large collective atom coupling, and the majority of atoms being in the ground state. However, the scheme is passive rather than active, and requires a separate probing laser to collect the signal. This second approach has the advantage of potentially scattering $\ll 1$ photons per atom, thus allowing squeezing in the system.

Our approach of using excited-state dependent population detection means that it can be more agile in its applications. We could perform active measurements of the B-field by inducing simultaneous emission from stretched Zeeman states, and take advantage of the

information contained in the beat-note of the emitted signal with reference light. So far we do not take advantage of information contained in the spectrum of the emitted light.

A disadvantage to this technique is the overhead of an appropriately chosen external readout cavity, which could present experimental challenges such as reduced optical access. The excitation method we use here requires phase-coherence between the excitation laser and the atoms along an axis orthogonal to the cavity mode, which is atypical for such setups. Finally, the readout time is not flexible but is self-initiated immediately after excitation of the ensemble above the equator of the collective Bloch sphere.

We emphasize that this study has not been exhaustive and further optimization would lead to a more definitive list of the quantitative advantages and disadvantages of our technique.

To address a comparison with other methods we have added the following sentences to the Conclusion on page 12: *“A SR readout scheme presents an active alternative to nondestructive measurement methods, where a single or multiple transitions can be simultaneously interrogated, and results in near-perfect collection efficiency of the emitted photons. This measurement method can be used in optical clock systems or sensors to obtain fast readouts with less than one scattered photon per atom.”*

Also, we added the following text: *“The higher duty cycle could be particularly beneficial for transportable optical clocks, where the fractional instability is limited by the performance of the clock laser and the associated aliasing of laser noise [39]. Compared to other methods developed to overcome dead-time effects, such as interleaving interrogation of two independent clocks [40], imaging in tweezer arrays [41, 42], and non-destructive dispersive probing [9], this method uses no secondary vacuum chamber, high numerical aperture lenses, or extra laser frequencies. It only requires a readout cavity and is compatible with existing cavities used for spin-squeezing [43].”*

(2) How does this readout scheme compare to straightforward atomic state detection in free space (it is asserted in the abstract that this method has “high-sensitivity” and later, in the introduction, that the cavity “drastically increases detection efficiency compared to free-space” fluorescence detection, but this claim is not clearly substantiated in the body of the manuscript)? The frequency discrimination implied by the error bars in figures 3a and 4 does not appear especially precise, and in any event should be compared quantitatively to standard readout with the same resources (e.g. atom number) in order to substantiate a claim of increased efficiency and/or sensitivity.

The method we present is inherently simple, leveraging no additional lasers for the readout and solely utilizing the photons provided by the interrogation laser, which represents the minimal possible heating. In straightforward state detection methods such as electron

shelving, an additional, spectrally broad, laser is applied which scatters many photons per atom.

Our statement that the approach “drastically increases detection efficiency compared to free space” was meant to refer to an enhancement in photon collection efficiency. In free space detection, photons scatter into a solid angle 4π , of which only a small fraction hits the detector. Whereas in our system, the scattered photons are about $1e4$ times more likely to exit through the cavity mode than into free space. By placing a detector behind a cavity mirror, the detection efficiency is therefore increased relative to free-space methods. We have revised this sentence on page 2 for clarity: “*This preferred directional emission allows for efficient collection of nearly all the signal photons compared to more straightforward atomic state detection in which most of the photons are lost into free space. Furthermore, we do not require any additional lasers as in electron shelving, since our readout relies on detecting photons directly from the clock interrogation.*”

The aim of the “high sensitivity” phrase was also to describe the high sensitivity per photon readout. We acknowledge this was not articulated clearly, and we thank the Referee for pointing out that this term is commonly associated with the SNR of the slope of a spectroscopic signal which we do not substantiate in our figures. We have adjusted the language in the abstract to instead read: “*highly directional emission of signal photons*”. Investigating the SNR of the technique and the achievable sensitivity of this readout scheme is the subject of current investigations in our lab.

(3) Instead, my understanding of the key result is that the scheme enables repeated highly non-destructive readout in less than two milliseconds, dominated by the 1.7 ms of cooling. Indeed non-destructive readout in free space often takes at least 10 ms or longer. Furthermore, the actual detection time is only a few microseconds. Given the relatively short excited state lifetime used in this work, this is a key enabling advantage. Yet this context is not clearly presented.

We appreciate the Referee's recognition of our scheme's capability for rapid and highly non-destructive readout, a distinct feature of our approach. We agree that the expedited state readout enabled by the enhanced emission rate—substantially faster than the natural lifetime—is a noteworthy advantage. Nevertheless, we contend that the mapping of the excitation threshold for a transversely excited atomic ensemble, along with the characterization of the novel Ramsey lineshape, stands as a central achievement of our study.

To emphasize the practical implications of our findings, we have elaborated in the conclusion on page 7 on the increased duty cycle and added the sentence: “*The higher duty cycle could be particularly beneficial for transportable optical clocks, where the fractional*

instability is limited by the performance of the clock laser and the associated aliasing of laser noise [39].”

(4) Some discussion of the limitations of this scheme should occur. For instance, more complicated sensing schemes involving Hahn echos or various dynamical decoupling sequences may not be possible since the atomic spin may become too inverted. Also, the inversion region for phase estimation is reduced in this readout scheme since all population fractions below threshold yield zero signal.

It should be possible to apply more complicated schemes, such as spin echoes and dynamical decoupling sequences, if we use the correct phase of the pulses such that the collective Bloch vector always navigates through the southern hemisphere of the Bloch sphere. Furthermore, sequences in which the Bloch vector passes above the equator should be possible if these sequences can be done much faster than the superradiant decay rate.

We have included two additional sentences to address these points in the conclusion on page 12: *“Other operations such as spin echoes should be possible without initiating collective cavity decay if these pulses are applied with the appropriate phases, such that the collective Bloch vector remains below the equator. Torquing the collective Bloch vector above the equator is permissible if it is done with durations much shorter than the inverse collective Rabi frequency because that is the rate at which atoms will start to superradiantly emit once above threshold.”*

Detailed comments

(5) Various quantities are vague and should be stated precisely, with error bars.

a. Cavity decay rate of 780 kHz lacks an error bar.

We thank the Referee for this comment and have adjusted this measured value in the text to include an error bar on page 3: *“780(4) kHz”*.

b. Why is a firm number for the finesse not provided? Cavity linewidth and free spectral range are both easily measured.

We have updated the text to provide a number with an error bar for the finesse: *“1001(5)”*.

c. Stated atom numbers presumably have a significant error bar which should be included. Typically atom number varies

Yes, the atom number typically varies with a standard deviation of about 10%. We have included an error bar on the stated atom numbers within the manuscript.

d. Normal mode splitting is said to be “highly resolved” but the data supporting this statement are not shown. At the very least the fitted $\sqrt{N}g$ with error bar should be provided. This data is later referenced to provide an expectation for the single-atom-single-photon coupling g . This further emphasizes the need for error bars! The simulated g and fitted g may or may not be in conflict with each other if proper error analysis is performed.

We have now changed the text to read: “A normal mode splitting measurement of $2g\sqrt{N} = 2\pi \times 5.42(14)$ MHz”. This value was derived from 30 repeated normal mode splitting measurements with $40(4)e6$ atoms as measured with absorption imaging, with error bars showing the standard deviation.

This comment helped us realize that using the delay time to derive the coupling strength, g , is not optimal as the underlying model neglects some important effects. As a corrective measure, we have standardized the value of g for all simulations throughout the paper, basing it on our robust normal mode splitting measurement. For the time delay plot depicted in Fig. 2(c), we fit a constant time delay offset of $437(17)$ ns to compensate for our non-zero velocity distribution and Doppler dephasing effects. These are not included in the theoretical model, but have been seen to cause delays on the order of a fraction of the pulse duration in Ref [17].

We thank the Referee for this comment and feel that the consistent use of g , as derived from our normal mode splitting measurement, strengthens our argument.

(6) Imperfect π -pulses are referenced as the primary limitation for the maximum inversion and therefore the maximum detected signal.

a. How much of a reduction do you see? How much of that can be attributed to excited state decay during the excitation time?

From measurements of the ground-state population via fluorescence on 1P1 immediately after a nominal π -pulse, we maximally excite approximately 80 - 90% of the atoms in our second-stage (red) MOT to 3P1. With a π -pulse duration of 600 ns, and by rigorously checking the time delays with a photodetector just outside the vacuum chamber, we ensure that the superradiant emission occurs after the excitation pulse has ended. With the transition's natural lifetime of 21 μ s, in 600 ns about 3% of the atoms would have decayed from the fully excited state. However, the decay while they are being excited would be less, around 1.5%.

We have added the following sentence in Section 3 page 4: “*With a π -pulse duration of 600 ns, we expect that maximally 2% of the atoms decay spontaneously during the excitation pulse.*”

b. From the statement “Slight deviations from a π -pulse always result in lower peak emitted intensity” I infer that your model is temporal intensity noise causing the Rabi frequency to fluctuate between pulses. If so, this is a slow variation which can be measured – is the observed reduction in π -pulse contrast consistent with these intensity fluctuations?

To clarify, we do not perceive the lower peak intensities near a π -pulse as a slow variation. The pump pulse intensity can be actively monitored through the 0th order of the pump pulse AOM. Our laser feedback does exhibit servo bumps at 1 MHz, which introduces fast phase noise at a relevant timescale for the π -pulses which have durations of $\sim 1 \mu\text{s}$.

While variations in pump pulses targeted at 50% fractional excitation might result in fluctuating excitation levels both above and below the intended target, aiming at 100% excitation inherently skews deviations towards lower excitations. This phenomenon largely accounts for the observed reduction in peaks in proximity to π -pulses. For further clarity, we revised the statement on page 4 to read: “*Due to imperfections in the excitation pulses such as an inhomogeneous beam profile and intensity fluctuations, measured peak powers near the maximum tend to be biased towards lower values. When targeting excitations below the maximum level, imperfections lead to symmetric deviations in both the peak amplitude and the integrated area of the superradiant emissions. However, in the case of aiming for complete excitation, such as with π -pulses, any imperfections will result in a reduced population inversion.*”

c. Or perhaps the model is different. If this is the case, what is the model and is the observed π -pulse fidelity consistent with experimental sources of those errors.

We have now added supplemental material describing the model, which assumes a spatially uniform Rabi frequency when the atoms are driven transversely by a noiseless laser. In the experiment, we have an uneven transmittance of our excitation beam caused by built-up strontium coating on the vacuum chamber windows. This results in a non-uniform pump intensity profile which causes spatially varying Rabi frequencies across the atomic ensemble. Furthermore, we have some laser noise around 1 MHz, as hinted by spectral analysis of the feedback. With these considerations, a maximal excitation fraction of 80-90% is reasonable.

(7) The authors have used a simulation to provide a theory curve in figures 2 and 3 and to provide comparison numbers in the main text. This simulation is not described anywhere. The contents and details of this model must be stated (what are the Hamiltonian and

Lindblad terms, what parameters are fixed by which calibration, what parameters are left as free parameters to be fit, etc.).

We have now included supplemental material detailing the model, including the Hamiltonian and Lindblad terms. The added material also describes the assumptions and parameters. Furthermore, we added two code examples: one to reproduce the simulation results and one to show the equations describing our system.

(8) I am confused by the sentence “Immediately after the SR pulse is emitted, the atoms retain an excited state population of ...”. This sentence does not appear to serve a purpose. Also, as defined earlier, $\langle \sigma^{22} \rangle_{t=0}$ is the excited state population at the end of the pump pulse. As mentioned earlier in the paragraph, there is some decay between the end of the pump pulse and the superradiant pulse so I would expect the excited state population after this decay to play a role rather than the original excited state population. But, regardless, I would further expect the excited state fraction after the superradiant pulse to be simply zero. ****except that atoms in ground state can absorb some of the superradiant pulse****”

We thank the Referee for the comment and recognize the importance of elucidating this point for the reader. The excited state fraction after a superradiant pulse does not always go to zero, except in the case of a fully inverted initial sample, or an excitation phase coherent with respect to the decay mode. To clarify, if the initial excited state fraction is 80%, the sample decays superradiantly only to 20% excitation, with the remainder decaying spontaneously outside of the cavity mode for long wait times.

A detailed explanation of this phenomenon, alongside corresponding figures, can be found in Ref. [21] (Fig. 2(b) and Fig. 4). We have added to this sentence in the text in Section 3, page 5: *“Immediately after the SR pulse is emitted, the atoms retain an excited state population of $1 - \langle \sigma^{22} \rangle_{t=0}$, leaving them in a subradiant state with respect to the cavity mode. The atoms which are left in the excited state after the SR pulse will then decay spontaneously into free space.”*

(9) The sentences beginning with and following, “Notably, the collectively enhanced lineshape exhibits a distinct kink...” argues that the threshold for superradiant emission would provide a good narrow error signal for locking a laser to atoms. This is a surprising argument since the emission is uniformly zero for all phase deviations below this threshold, so this error signal would provide no information as to whether the laser is exactly at the lock point or significantly off in one direction. Indeed the error signal proposed, the frequency locator, operates much more conventionally where the signal is large and with a high sensitivity to small changes in the laser phase.

We agree that for phase deviations below the threshold, the error signal is flat, providing no information, and that this warrants further discussion. We have added a supplemental note regarding this, and in the referred-to sentence, we have added: “(*Supplementary Information, Section 2*)”.

Typically in a Ramsey scheme, the frequency or phase is stepped around locations of maximal slope. The highest slope in the collectively enhanced lineshape is just above the kink. If we alternated interrogations between the two kinks of one fringe, even if one step yielded no pulse, the other would have a maximally increased pulse amplitude. In this way, even if one step is out in a flat region that yields no photons, in pairs of interrogations, there can still be a useful feedback signal from the FL. Here the FL slope is very steep, - however there is only a small dynamic range with respect to laser detuning. This is demonstrated by the simulation using frequency steps of 0.5 FRR above in Fig. 1, which is now in the supplementary material.

(10) The sentence “Also, it is possible to get 100s of useful pulses...” is colloquial and vague. What is meant by useful? The data in figure 4a appears to show almost a factor of three reduction in atom number after 100 (not 100s) repeated pulses. It is not clear what atom number I should compare to.

We thank the Referee for highlighting the imprecision of our original wording. We acknowledge that the sentence regarding the utility of the pulses is colloquial and potentially ambiguous. In light of this, we have opted to remove the sentence from the text.

(11) How was the frequency locator detuning of 0.1 FRR chosen? Is this optimized to provide the largest derivative at zero laser detuning?

Our aim was to demonstrate a clear alternating pattern of the resulting superradiant pulse amplitudes when stepping the frequency about a fringe. Our particular choice of 0.1 FRR was experimentally chosen to ensure reasonable signal size for the high/low pulse amplitudes. Traditionally, the frequency or phase of the interrogation laser is stepped between the steepest slopes on a spectroscopic fringe, near 50% excitation. If future investigations can improve upon the SNR just above the kinks, a larger stepping size compared to the FRR would provide a steeper slope and thus a higher sensitivity to laser detuning. We demonstrate this by simulating the FL shapes for various stepping sizes in the added Supplementary Information, Section 2. Currently, we are limited by the SNR and contrast of our fringes and therefore decided to remain closer to the center of the fringe.

To address this, we have added the following sentence on page 7: “*The chosen step size represents a trade-off between maximizing dynamic range and optimizing SNR of our fringes. In a sensor, the ideal step size will be chosen to conform with the requirements and stability of the system (Supplementary Information, Section 2).*”

(12) In the outlook, it is suggested that the readout scheme could be applied to the clock transition in Yb or Sr, which could yield advantages in reducing Dick effect noise. I agree that this could be valuable, especially for many-times-repeated short interrogation time clocks. It is worth noting that there has been work to minimize Dick effect noise in optical lattice clocks (e.g. <https://arxiv.org/abs/1305.5869>) and tweezer clocks (e.g. <https://arxiv.org/pdf/1904.10934.pdf> and <https://arxiv.org/pdf/1811.06014.pdf>). It might be useful to include some comparative statement with alternative methods to reduce this source of noise.

We thank the Referee for commenting on the potential capabilities of reducing Dick effect noise. We agree such a comparative statement is useful, as each method has its unique advantages and disadvantages. We have included the following statement in the conclusion on page 12: *“Compared to other methods developed to overcome dead-time effects, such as interleaving interrogation of two independent clocks [40], imaging in tweezer arrays [41, 42], and non-destructive dispersive probing [9], this method uses no secondary vacuum chamber, high numerical aperture lenses, or extra laser frequencies. It only requires a readout cavity and is compatible with existing cavities used for spin-squeezing [43].”*

Reviewer #1 (Remarks to the Author):

The authors have substantially improved the manuscript to satisfactorily address the points raised in the last review stage. I recommend publication without further change.

Reviewer #2 (Remarks to the Author):

The authors have clarified my main concern and addressed the other comments as well. I think the manuscript is suitable for publication in Nature Communication

Reviewer #3 (Remarks to the Author):

Bohr et al have significantly revised their manuscript and directly addressed the key issues raised by all reviewers in the original manuscript.

There is one detailed point I still do not understand, which was not addressed in the response to detailed comment (8). Namely, why is the remaining population given by $1 - \langle \sigma^{22} \rangle_{t=0}$, where $\langle \sigma^{22} \rangle_{t=0}$ has been defined as "state population at the end of the pump pulse", as opposed to the excited state population at the beginning of the SR pulse. As mentioned earlier in that paragraph, there is significant decay between the end of the pump pulse and beginning of the SR pulse so I would think that this decay fraction should affect this population that remains in the excited state. I hope that the authors clarify this point in the final manuscript.

Nevertheless, in its current form I can recommend publication of this manuscript.

REVIEWERS' COMMENTS

Reviewer #1 (Remarks to the Author):

The authors have substantially improved the manuscript to satisfactorily address the points raised in the last review stage. I recommend publication without further change.

We thank the Referee for the positive feedback and recommendation for publication.

Reviewer #2 (Remarks to the Author):

The authors have clarified my main concern and addressed the other comments as well. I think the manuscript is suitable for publication in Nature Communication

We thank the Referee for the positive feedback and recognition of the manuscript's suitability in Nature Communications.

Reviewer #3 (Remarks to the Author):

Bohr et al have significantly revised their manuscript and directly addressed the key issues raised by all reviewers in the original manuscript.

There is one detailed point I still do not understand, which was not addressed in the response to detailed comment (8). Namely, why is the remaining population given by $1 - \langle \sigma^{22} \rangle_{t=0}$, where $\langle \sigma^{22} \rangle_{t=0}$ has been defined as “state population at the end of the pump pulse”, as opposed to the excited state population at the beginning of the SR pulse. As mentioned earlier in that paragraph, there is significant decay between the end of the pump pulse and beginning of the SR pulse so I would think that this decay fraction should affect this population that remains in the excited state. I hope that the authors clarify this point in the final manuscript.

Nevertheless, in its current form I can recommend publication of this manuscript.

We thank the Referee for their positive feedback on our revised manuscript.

We apologize for not addressing this query in the previous response. The Referee is indeed correct in pointing out that the remaining population would be affected by the decay in the interval between the end of the pump pulse and the onset of the superradiant emission. We have revised the referred-to sentence (“Immediately after...”) as follows: “*After the SR pulse is emitted, the remaining atoms in the excited state will be subradiant with respect to the cavity mode (in the case of no spontaneous emission, this population is equal to \$1 - \langle \sigma^{22} \rangle_{t=0}\$, as described in cite{Hotter2023}.*”